# The Impact of the COVID-19 Pandemic on ICU Healthcare Professionals: A Mixed Methods Study

**DOI:** 10.3390/ijerph18179243

**Published:** 2021-09-01

**Authors:** Cristina Moreno-Mulet, Noemí Sansó, Alba Carrero-Planells, Camelia López-Deflory, Laura Galiana, Patricia García-Pazo, Maria Magdalena Borràs-Mateu, Margalida Miró-Bonet

**Affiliations:** 1Department of Nursing and Physiotherapy, University of Balearic Islands, 07122 Palma, Spain; cristina.moreno@uib.es (C.M.-M.); alba.carrero@uib.es (A.C.-P.); camelia.lopez@uib.es (C.L.-D.); patricia.garcia@uib.es (P.G.-P.); maria-magdalena.borras1@estudiant.uib.cat (M.M.B.-M.); mmiro@uib.es (M.M.-B.); 2Balearic Islands Health Research Institute (IDISBA), 07120 Palma, Spain; 3Department of Methodology for the Behavioral Sciences, University of Valencia, 46003 Valencia, Spain; Laura.Galiana@uv.es

**Keywords:** COVID-19, mixed methods, ICU healthcare professionals, moral distress, professional quality of life

## Abstract

The large numbers of patients admitted to intensive care units due to COVID-19 has had a major impact on healthcare professionals. The incidence of mental health disorders among these professionals has increased considerably and their professional quality of life has suffered during the pandemic. This study aims to explore the impact of the provision of COVID-19 patient care on ICU healthcare professionals. A mixed methods study with an exploratory concurrent design was conducted between June and November 2020 in the Balearic Islands, Spain. Data were collected using a self-report online survey (*n* = 122) based on three validated questionnaires, and individual semi-structured in-depth online interviews (*n* = 11). Respondents scored 2.5 out of 5 on the moral distress scale, moderate/high on the compassion satisfaction scale, and moderate on the burnout and compassion fatigue subscales. Age was significantly and negatively related to professional quality of life but was positively related to workload and unavailability of protective equipment. Three main groups of themes relating to the impact of the pandemic emerged from the in-depth interviews: (a) clinical, (b) professional, and (c) personal and family impacts in the two waves. ICU healthcare professionals should be viewed as second victims of the COVID-19 pandemic as they have suffered significant psychological, professional, and moral harm.

## 1. Introduction

On 11 March 2020, the World Health Organization declared COVID-19 a global pandemic [1]. In Spain, a state of emergency was declared three days later: fundamental rights were limited, and lockdown restrictions imposed on the entire population. This situation lasted until May 2020, marking the first wave of the pandemic. After two months of low infection rates and the easing of restrictions to reopen the economy, the second wave began in August 2020 and lasted until October 2020, this time without stay-at-home lockdowns.

With a total of 47 million inhabitants, an estimated 1.5 million people were infected in Spain, with 42,000 deaths between the first and second waves.

Health professionals have been one of the most affected groups in all countries with a high incidence of infections [2]. In Spain, during the first wave of the pandemic, the percentage of infected healthcare professionals ranged from 25% to 30% in the case of physicians, from 45% to 55% in the case of nurses and nursing assistants, and from 5% to 10% in the case of orderlies [3]. Until the beginning of May 2020, 40,961 cases of COVID-19 were reported in healthcare professionals (76.5% were women). This number of cases represented 24.1% of the total number of COVID-19 cases declared in Spain until then. According to [4], 10.5% of the reported cases in healthcare professionals needed to be hospitalized, 16.2% developed pneumonia, 1.1% were admitted to hospital intensive care units (ICUs), and 52 healthcare professionals died from COVID-19. Among the general population, 10% of patients were admitted to ICUs, putting additional pressure on hospitals, especially on these units, and increasing the need for human and material resources. In the Balearic Islands specifically, 98 hospital ICU beds were made available, 84 of which were designated for COVID-19 patients, compared to the usual 60 ICU beds.

The increase of the welfare activity and the limited number of ICU beds equipped with ventilators were accompanied by the emergence of ethical conflicts related to the need to carry out a clinical triage to make rationing decisions on which patients would be admitted and offered life support on the basis of their age and their level of chronicity or fragility [5,6,7]. These ethical decisions were at risk to cause moral injury to patients, families, and healthcare professionals [7].

Moral distress arises when one knows the ethically appropriate action to take, but the internal barriers (ones’ values) and the external barriers (institutional constraints such as the lack of resources) make it nearly impossible to pursue the right course of action [8,9]. In this sense, the American Association of Critical Care Nurses [8] identifies some factors that can cause moral distress such as drastic increases in workloads and precarious working conditions. Moral distress is mostly experienced by nurses, since it mainly appears in those professionals whose identities are intrinsically linked to altruism and compassion values [10].

Due to their characteristics and working conditions, ICUs have always been considered a place where healthcare professionals are at high risk of experiencing moral distress [11,12]. During the pandemic, professionals have suffered moral distress as a result of their exposure to potentially harmful situations such as repeated empathic commitment to patients grief and loss [13], limited access to proper personal protective equipment [14,15], poor perception of organizational support [16,17], worries and concerns regarding becoming infected and infecting their family members [15,18], uncertainty about disease containment strategies [14], and concerns about seeing patients die [18]. Some of these situations could be understood as institutional violence or inappropriate ethical climates [19,20].

In addition, there is a relationship between moral distress and professional quality of life [21,22], specifically with burnout syndrome [22] and compassion fatigue [23]. Burnout Syndrome (SBO) is a psychological state emerging as a prolonged response to chronic interpersonal stressors on the job [24]. Under normal circumstances, healthcare professionals may seek comfort in family and social life to cope with occupational stress, but during the COVID-19 pandemic this was no longer an option. Hence, the exhaustion of healthcare professionals due to emotional distress has become a growing concern during the pandemic [6].

However, occupational stress and SBO are insufficient to understand the professional quality of life [25,26]. Thus, a reference must be made to compassion fatigue (CF). CF is defined as a stress condition resulting from helping or wanting to help people suffering from some type of trauma [27]. Specifically, what is proposed is that continuous exposure to traumatized people produces tension and chronic worry, resulting in potential psychological difficulties [25], physical and emotional exhaustion [28], inability to feel empathy and compassion toward patients [29], and lower resilience to others’ suffering [25,30].

In contrast to CF, and from this same approach, compassion satisfaction (CS) is defined as the gratification resulting from exposure to traumatic events [31] or as the level of enjoyment resulting from helping others [26]. Stamm [26] coined the term “quality of professional life” to cover CF, SBO, and CS processes.

During the pandemic, the incidence of anxiety disorders, depression, and post-traumatic stress disorder increased considerably among ICU professionals during the pandemic [16,32,33,34,35,36,37,38,39,40,41,42,43,44]. Professional quality of life has been undermined by an increase in burnout syndrome and compassion fatigue, especially among women and nurses [40,45,46,47,48,49,50]. Regenold and Vindrola-Padros [51] argue that gender is significant when understanding the experiences of healthcare professionals during COVID-19 as it illuminates ingrained inequalities and asymmetrical power relations, with gendered organizational structures interacting to shape healthcare workers’ experiences. Differences by gender and age have previously been found in health care professionals’ quality of life, as shown by the fact that women present higher values in compassion fatigue [52,53,54], furthermore, age is also associated to professional quality of life, according to El-Shafei et al. [55], the older the professional, the greater the compassion satisfaction.

Contributing factors include: (1) the lack of material resources (personal protective equipment—PPE, hospital beds, and ventilators), (2) shortage of nurses, (3) worsening working conditions for healthcare professionals involving intense working hours and extreme workloads, and (4) clinical/health consequences of changing environments, increased adverse events, and clinical complications [6,36,37,49,56,57,58]. Other studies point to concerns among experienced ICU professionals about unskilled junior professionals joining their units [58,59,60]. ICU professionals have also faced ethical dilemmas and conflicts arising from the need to establish more restrictive admission profiles and the inability to provide emotional support and information to families in person [7,32,58,61,62,63]. To address this situation, a number of studies suggest the implementation of organizational coping strategies such as improving human resources policies, designing targeted psychological interventions for professionals, and having ethical guidelines for difficult decisions in place in ICUs [6,7,13,34,37,64,65,66,67].

It is therefore imperative to analyze the impact of the COVID-19 pandemic on healthcare professionals caring for COVID-19 patients in order to establish strategies to minimize and systematize the consequences of similar health crises and shocks in the future.

The aim of this study was to explore the impact of care provision at COVID-19 ICUs on healthcare professionals working at public hospitals in the Balearic Islands, taking their gender and professional category into consideration.

## 2. Materials and Methods

This is a mixed methods study using an exploratory concurrent design [68].

The study was carried out at the six ICUs at the public hospitals in the Balearic Health Service. The study population numbered 800 professionals (98 critical care physicians, 417 nurses, and 285 nursing assistants). For both the quantitative and qualitative parts of the study, the inclusion criterion for participants was having more than two weeks’ experience working at a COVID-19 ICU.

### 2.1. Quantitative Design

The quantitative design was a descriptive cross-sectional study that was carried out using a self-report online survey assessing professional quality of life. The survey was created using the SurveyMonkey platform, which included a series of sociodemographic data, information on workplace characteristics and availability of protective equipment, as well as the following instruments:

The short version of the Professional Quality of Life (Short-ProQOL) scale [69], a shortened version of Stamm’s ProQOL scale [26] measuring professional quality of life. The questionnaire contains three subscales (compassion satisfaction, compassion fatigue, and burnout) consisting of 3 items each, rated on a scale ranging from 0 (never) to 5 (always). It has previously been used in professionals working with high emotional demands [70,71,72]. In our sample, internal consistency estimates were adequate (*α* = 0.84 for compassion satisfaction; *α* = 0.68 for burnout; and *α* = 0.84 for compassion fatigue).

The Moral Distress Scale–Revised (MDS-R) [73] measures moral distress in certain situations. Respondents indicate the frequency and levels of distress they experience when faced with a stressful situation. The MDS-R contains 21 items rated on a Likert scale ranging from 0 to 4 with each item measuring 2 aspects: how often the stressful situation arises and the intensity or level of distress it causes. Reliability was 0.92.

The Professional Self-Care Scale (PSCS) [74] was used to assess self-care via three subscales or dimensions: physical, psychological, and social self-care. The PSCS contains 9 items rated on a 5-point Likert scale ranging from 1 (completely disagree) to 5 (completely agree). The instrument was originally developed and validated for a sample of palliative care professionals and has previously been used in similar contexts [70,72,74]. Reliability estimates were 0.70 for physical self-care, 0.81 for psychological self-care, and 0.60 for social self-care.

Participants were recruited for this phase by emailing hospital and ICU managers and disseminating the survey via WhatsApp groups. A total of 122 professionals completed the survey. The response rate was 15%.

The data were analyzed via a descriptive analysis of the characteristics of the sample and an inferential analysis of the study variables. The statistical significance threshold was set at *p* < 0.05. SPSS 25.0 for Windows (IBM, Armonk, NY, USA) was used for quantitative analysis.

### 2.2. Qualitative Design

The qualitative design employed a purposive sampling method. The researchers interviewed 11 critical care professionals using individual online semi-structured in-depth interviews between June 2020 and November 2020 [75]. Pre-interviews were conducted prior to the formal interviews to ensure the rationality of the interview structure and the representativeness of the subjects [62,76]. The data were saturated with 11 interviews [77,78]. The participants were 4 intensive care nurses, 4 intensive care doctors, and 3 nursing assistants. Participant profiles were devised based on profession and gender. The sample was balanced in proportion to the number of professionals in each ICU.

Participants were recruited using a snowball strategy. Interviews were carried out by two skilled researchers. The principal investigator carried out all the interviews and there was always another researcher who observed and took field notes. Each interview lasted between 60 and 90 min.

The main questions guiding the interviews were as follows:What was your experience in providing healthcare during the COVID-19 crisis?Were there any problems or conflicts arising from that situation? Could you give some examples? Do you remember any particularly difficult moments? In what sense?How has this affected you personally and professionally? What have you been most concerned about, and do you think it will have future implications?

The interviews were audiotaped and then transcribed verbatim. The conventional method of qualitative content analysis was used, adopting a primarily inductive or data-driven approach [79]. The analysis was conducted from the beginning of the data collection process through to January 2021. Interviews and observation notes were codified independently by four researchers. Once finished, all researchers met to compare their results. Where their codifications differed, researchers explained their thinking processes. Through a process of dialogue and comparison, they reached an agreement on the coding system. Therefore, the codification of each interview and observation notes was the result of five codifications: four independent ones and a joint one. Once the list of codes had been completed, two team members drew up the analysis subcategories and categories and revised the codes under each of them. Afterwards, they compared the coherence of each code and revised the list of codes, excluding the ones that lacked a specific sense and unifying the ones that, although having the same meaning, had been codified with different codes. It is important to note that few codes referring to different categories were unified and that most of the fusions of codes were produced in the same categories. The researchers—registered nurses with clinical experience in ICU—provided a rich description of the research context, selection of participants, and data collection and analysis process to enhance the study’s transferability.

The study protocol was approved by the Research Ethics Committee at the University of the Balearic Islands (Ethics code: 152CER20). After approval, the participants were informed about the study objectives and reassured that their data would be kept confidential at all times and that they were free to withdraw from the study at any time without giving a reason. The participants signed an individual informed consent form.

## 3. Results

### 3.1. Quantitative Results

#### 3.1.1. Characteristics of the Participants

A total of 122 professionals (65% nurses, 24% nursing assistants, and 10% physicians) completed the questionnaires. Among them, 74% felt that they had the necessary protective equipment, 89% saw their workload increase, and 90% reported caring for patients who had died from COVID-19. Other demographic characteristics are shown in Table 1.

#### 3.1.2. Description of Psychological and Moral Status

Table 2 describes the results for each of the psychological and moral variables. Moral distress levels were on the lower-middle side of the scale (2.5 on a scale from 1 to 5). In terms of professional quality of life, a mean of 40.4 points was obtained for compassion satisfaction, representing a moderate-to-high level. Compassion fatigue and burnout obtained mean scores of 26.5 and 27.5 respectively, representing moderate levels.

#### 3.1.3. Relation between Socio-Demographics and Psychological and Moral Status

In relation to the factors influencing professional quality of life, age and the risk of experiencing burnout were significantly and negatively related (*r* = −0.186; *p* = 0.048), indicating that older professionals experienced lower levels of burnout. On the other hand, there was no evidence indicating significant differences in moral distress between two genders (*t* (98) = −0.787; *p* = 0.433). The multivariate analysis of variance (MANOVA) exploring gender differences in professional quality of life failed to identify statistically significant differences (*F* (3.107) = 2.563; *p* = 0.059; *η2* = 0.067). Follow-up ANOVAs, however, suggested differences in compassion satisfaction in favor of male respondents (Table 3 and Table 4). As for the comparison between the various professional disciplines studied, the MANOVA was not statistically significant (*F* (6.212) = 1.695; *p* = 0.124; *η2* = 0.046), but the post-hoc tests revealed a higher presence of compassion fatigue among nursing assistants compared to nurses (*p* = 0.048). The other comparisons showed no statistically significant differences regarding the type of contract (permanent or temporary) or workplace. Analysis of variables relating to the direct impact of COVID-19 failed to show that undergoing preventive quarantine or working with infected co-workers influenced levels of professional quality of life or moral distress.

Increased workloads were not related to moral distress levels (*r* = −0.048; *p* = 0.636), but they were related to professional quality of life, with higher compassion fatigue (*r* = 0.234; *p* = 0.013) and burnout scores (*r* = 0.193; *p* = 0.041) and lower compassion satisfaction scores (*r* = −0.198; *p* = 0.036).

#### 3.1.4. Relation between COVID-19 Related Variables and Psychological and Moral Status

Analysis of variables pointed no statistically significant relation between undergoing preventive quarantine or working with infected co-workers and professional quality of life or moral distress.

However, the availability of protective equipment was significantly related to professional quality of life in the study sample (*F* (3.105) = 3.191; *p* = 0.027; *η2* = 0.084). Specifically, professionals who lacked sufficient protective equipment experienced lower levels of compassion satisfaction and higher levels of burnout, resulting in poorer professional quality of life and higher levels of moral distress (*t* (96) = 3.250; *p* = 0.002). Further details can be found in Table 3 and Table 4.

In addition, the perception among professionals of being able to properly support patients during their end-of-life process was statistically significant related to compassion satisfaction, with higher compassion satisfaction means for those who provided patient support. By contrast, no statistically significant differences in moral distress were found between those who were and were not able to provide patient support during the end-of-life process *t* (90) = 0.694; *p* = 0.078). Descriptive statistics are offered in Table 4.

### 3.2. Qualitative Results

Table 5 shows the profiles of the 11 professionals who were interviewed. Three major themes were extracted from the data analysis: clinical impact, professional impact, and personal/family impact on intensive care nurses, intensive care doctors, and nursing assistants. Table 6 summarizes the themes, sub-themes, and illustrative quotes from the qualitative data obtained during the two waves. The narratives of these professionals provided insight into the differences between these types of impacts in the two waves. The results are reported chronologically.

#### 3.2.1. First Wave in Spain (March–June 2020)

##### Clinical Impact

Professionals perceived the onset of the pandemic as unexpected and sudden, accompanied by highly intense feelings of uncertainty and disbelief (P2, N1).

On a clinical level, the number of admissions to ICUs grew exponentially, like other countries [2], forcing the number of hospital beds to be increased (N1), bringing normal surgical activity to a standstill, and increasing staffing ratios (P2). Some newly recruited professionals had no experience in the ICU and training them added to the workload of more experienced professionals (N4).

Their workloads increased considerably due to the clinical severity of the patients being admitted and complications linked to the use of PPE. The need to perform activities more quickly and efficiently than usual due to the risk of contagion, coupled with the heat, suffocation, and discomfort caused by wearing PPE, took a physical and psychological toll, especially on nurses and nursing assistants (P1). In connection to PPE, professionals sometimes prioritized clinical safety over personal safety (N2). They were initially concerned about the availability and suitability of PPE (P2), but PPE was eventually made available to them at all times, in contrast with other departments, to the extent that they began to consider themselves ‘the hospital elite’ (NA3).

In that context, nursing assistants in particular changed their way of working: they stopped having patients assigned to them and began to provide technical support to any professional requiring it. They perceived this new dynamic as provoking a decline in the quality of the care they were able to provide.

The mechanization of the work “like in an assembly plant” (NA3), the “dehumanization of care” (N1), the loneliness of patients, and the absence of families had an immense emotional and ethical impact on professionals (P2), especially with regard to end-of-life processes and the limitation of life support (N4, N1). They also recalled the improvement and recovery of a number of COVID-19 patients and the gratitude of their families with satisfaction (N3).

##### Professional Impact

Professionals reported major changes taking place at a professional level. Some nurses reported that, for the first time, they had heard colleagues considering “leaving the profession” (N4). However, colleagues’ relationships with one another, their ability to adapt, and their resilience, patience, humility, humanity, and empathy were also reinforced (NA2, N4). This enhanced team cohesion, interprofessional collaboration, and satisfaction (N1) blurred power relations between professional categories by prioritizing expertise over rank. They also felt that their work was recognized by patients’ relatives and by society as a whole.

##### Personal and Family Impact

The heavy workloads, extra shifts, and high levels of stress to which they were exposed led to physical fatigue, stress, and insomnia (N1, NA2). A number of the professionals suppressed their emotions at work and burst into tears when they arrived home (NA3).

They lived in fear of infecting their family members, especially the elderly and children (N4). As a precautionary measure, some professionals moved to special hotels provided by the government to protect their families or took their children to stay with their grandparents or other relatives. With respect to gender, women explained their emotions, distress, and concern for their families and the self-cleaning and self-disinfection “rituals” performed at home in more detail (N4), whereas men provided less information on this topic and rationalized the situation more (P1).

#### 3.2.2. Second Wave in Spain (August–November 2020)

##### Clinical Impact

In the second wave, although more information and protocols were available (N1), care provision for the growing number of COVID-19 patients had to be balanced with the usual care activity. Staffing in the ICU could not be reinforced, which increased the workload for professionals (NA3). At the same time, the number of professionals on sick leave or in quarantine increased, requiring some professionals to work extra shifts (N1). Professionals criticized the lack of foresight among managers and human resource teams in particular, whose policy was to rotate newly recruited professionals instead of retaining experienced ICU professionals (NA3).

In this wave, the availability of PPE ceased to be a problem for professionals, who became accustomed to working in these conditions (P4). Similarly, providing care to COVID-19 patients was easier due to increased knowledge about the disease, its transmission, and its clinical management (N1). As a result, professionals were less focused on the clinical impact of the second wave on patients than on its impact on their professional quality of life.

##### Professional Impact

In the second wave, professionals reported becoming increasingly burned out (N1). The resilience narratives from the first wave were replaced by references to burnout syndrome (NA3). The dynamism, dedication, willingness, and professional pride reported during the first wave were replaced by fatigue (N1), outrage, and helplessness as patients admitted to the ICUs for COVID-19 were increasingly younger and had been infected after contravening COVID-19 restrictions (P4). The supportive, comradely atmosphere sometimes became toxic, with non-stop complaints and burnout, which easily spread throughout the unit (NA3). Professionals criticized the lack of institutional psychological support (P4) and called for care plans specifically tailored to them.

##### Personal and Family Impact

Professionals’ fear of infecting the people they lived with gradually subsided, and they began to find ways to balance their work and family lives. However, they continued to comply with public health restrictions and recommendations, which prompted their outrage at the low levels of compliance among citizens (N1).

Their tiredness and insomnia worsened in the second wave, adversely affecting their quality of life. Professionals also recalled distressing and recurring experiences with some patients (P4).

## 4. Discussion

This study has explored the impact of the provision of COVID-19 patient care on intensive care nurses, intensive care doctors, and nursing assistants. Our findings show that these professionals faced a combination of clinical, professional, and personal circumstances that may have turned them into second victims of the pandemic. In addition, a number of the findings can be viewed as characteristic of unethical or institutionally violent settings [9,11,19,20,80].

During the first wave especially, professionals perceived a considerable increase in workload that was linked not only to the complexity of COVID-19 patient care [49,56,57,58], but also to the continuous modification of protocols and procedures and the need to train new professionals and professionals from other units who cancelled their clinical activity such as those who worked in operating rooms or in outpatient clinics [36,58,59,81]. This had a negative impact on professional quality of life, as suggested by previous studies [41].

At the beginning of the pandemic, the availability of PPE was one of the issues of greatest concern to professionals [33,35,36,81]. Our findings are in line with previous research showing that professionals who lacked sufficient PPE had a poorer professional quality of life and greater moral distress and ethical conflicts due to confusion, uncertainty, and ethical dilemmas arising from having to choose between providing safe care and protecting themselves and their families [58,61]. However, the ICU professionals in our study felt privileged, as unlike other units within their hospitals, they were sufficiently and adequately equipped. Like in other studies [82], participants in this study did not have high contagion rates despite working in COVID units.

Another clinical consequence was professionals’ perception of poorer quality patient care. The provision of care during the pandemic entailed the mechanization and Taylorization of care, which was experienced as the dehumanization of care practice [83]. Feelings of alienation among professionals took a moral toll on them [6,12]. Feelings of helplessness and moral distress arise from being unable to provide care as one would wish despite one’s best efforts [65]. This was particularly relevant among the female participants in this study due to issues linked to social and institutional gender expectations [84].

Evidence shows that clinical decisions, like establishing selection criteria for patients to be admitted to the ICU, is a source of legal and ethical dilemmas [7,58,61,85]. However, the participants in our study did not have to face this difficult decision, unlike other regions in Spain. Situations related to end-of-life processes had the greatest ethical impact on the professionals in our study. Many patients died and could not always be accompanied by their families. Professionals who were able to adequately support patients in their end-of-life processes experienced higher levels of compassion satisfaction [30]. This shows that ICU professionals have normalized family involvement and participation in the ICU, which have been encouraged in public hospitals for the last 10 years [62]. With the pandemic, there is a risk of a return to a rigid, compartmentalized model where families are excluded from ICUs.

From a professional perspective, the data show that participants had high levels of stress and moderate levels of compassion fatigue and burnout [45,49]. In line with other studies, younger professionals had higher levels of burnout than older ones [42,50]. On the other hand, the men in our study, regardless of their profession, had higher rates of compassion satisfaction than women, which is not consistent with previous studies [41,54]. When it comes to professional category, nursing assistants experienced the highest levels of compassion fatigue, perhaps due to gender-related issues, lower levels of expertise and training, limited access to formal sources of information, and lack of clinical knowledge and updated information on patients’ medical status [42,44].

Due to the harsh conditions of professional practice and the clinical situations experienced, some professionals considered leaving the profession, as observed in other countries [36]. Their exhaustion increased as the pandemic progressed, causing them to lose the resilience skills they had developed during the first wave. Consequently, in the second wave, feelings of outrage emerged among professionals at the insincere acknowledgement of their work by managers and politicians and the lack of healthcare and health prevention plans for professionals on the part of organizations [43,47,63].

Despite this, the participants also observed that pandemic care had had a positive impact on them as professionals. In the first wave, the development of coping strategies focusing on resilience and teamwork attitudes [6]—which are common to professionals exposed to crisis situations and closely linked to their commitment to patients and organizations—was imperative [67,86]. The need to constantly adapt to emergency situations demanded greater inter-professional cohesion and collaboration from professionals, which eventually resulted in a common identity that transcended their disciplines and them as individuals [58,60,67,86].

The pandemic also had an impact on professionals’ personal and family lives, negatively affecting their physical and emotional health [28]. Insomnia, fear of becoming vectors of transmission, and infecting their loved ones were the main consequences identified [13,16,36,37,38,41].

Finally, our findings highlight the need to implement care policies for professionals in similar health crises and shocks in the future, such as establishing psychological intervention programs for professionals on the frontline of care, developing ethical organizational cultures, re-humanizing care for patients and families, and ensuring safe environments for patients and professionals [34,37]. Specifically, we believe that nurses and nursing assistants need to specialize further in critical care and that stable human resources policies should be promoted [64].

The findings from this study also allow us to point out some educational and research implications. Due to pandemics or shock situations lack of prediction and control and due to the impacts these situations can have on professionals, we consider it is important to train professionals in expert and advanced skills in safety and risk management, and in particular, to train and mentor health professionals students in certain psychological resources, including resilience skills [87,88,89,90].

Finally, we consider it is important to point out some future research directions such as continuing to explore the differences and similarities of professionals’ clinical, professional and personal impact between different health services (such as emergency services, primary health care services), between genders and between years of experience (especially of those healthcare students who started working in the context of the pandemic), and also to explore the short and long-term impact of the pandemic on patients and their family members, especially on chronically ill patients and high dependency patients.

## 5. Limitations of the Study

The major weakness of this study is the small sample size, due to low response rate to the electronic survey, indicating particular difficulties in reaching the target sample via email. Possible explanations include the fact that not all employees use their work email addresses or that their workloads are so heavy that responding to a questionnaire is not viewed as a priority. This small sample size and response rate could affect the generalization of the results. Other shortcomings of current research include the absence of covariates that have been traditionally related to workers’ mental health, such as parental status, marital status, or social support outside the workplace; the fact that the set of measurements came from the same source, which could lead to the possibility of a common variance bias; the cross-sectional nature of the study; or the use of self-report measures for the quantitative part.

In the context of the COVID-19 pandemic, a decision guided by the existing legal restrictions on social gatherings, work overload of healthcare professionals and risk of contagion was made to only conduct individual interviews, setting aside to plan observational methods within ICUs settings. Thus, the sole use of conversational techniques allowed us to explore participants’ discourses and perceptions, but it did not allow us to contrast them with participants’ practices in the natural context.

On the other hand, rapport-building in researcher-participant relationships was challenged by the need to replace face-to-face interviews with online interviews (only one of them was conducted face-to-face at the request of the participant). However, engaging in virtual qualitative research made this study possible by reconciling restrictions on meetings with qualitative research methods [91,92].

The mixed methods design strengthens our results. The findings from the interviews and surveys pointed in the same direction. In addition, the qualitative interviews allowed us to explore differences between the first and second waves.

## 6. Conclusions

Critical care professionals may be regarded as second victims of the COVID-19 pandemic because of the enormous impact on their clinical, professional, and personal lives. Changes in care provision linked to the need to adapt to anti-COVID measures, increased workloads, and patient loneliness have negatively affected their professional quality of life, increasing their levels of compassion fatigue and burnout. Fortunately, the ICUs in the Balearic Islands were not faced with the ethical conflict of limiting patient admissions to the ICU and had sufficient and appropriate protective equipment, unlike other regions in Spain. The availability of protective equipment has proven critical given its potential impact on moral distress. Therefore, we may conclude that the perception of a safe environment is associated with lower burnout syndrome and moral distress. In addition, on a personal and family level, professionals suffered greatly from the fear of infecting their family members and changed the way they lived together during the first wave.

## Figures and Tables

**Table 1 ijerph-18-09243-t001:** Characteristics of the participants (*n* = 122).

Variables	Percentage	*M* (SD)
Age (years)		39/(9)
Sex		
Female	81.1%	
Male	18.9%	
Occupation		
Physicians	10%	
Nurses	65%	
Nursing assistants	25%	
Hospitals Hospital in Mallorca Hospital in Ibiza Hospital in Menorca	85% 11% 12%	
Type of contract		
Permanent	33%	
Locum	23%	
Temporary	44%	
Professional experience (years)		38/(13)
Family circumstances		
(1) Living alone	18%	
(2) Living with a partner	27%	
(1) or (2) with dependents	46%	
Have you had COVID-19?		
No	86%	
Yes	4%	
Has anyone in your family had COVID-19?		
No	87%	
Yes	13%	
Have you had to self-isolate?		
No	75%	
Yes	24%	
Has anyone in your department been infected with COVID-19?		
No	14%	
Yes	86%	
Has your unit provided you with protective equipment?		
No	25%	
Yes	74%	
Has your workload increased?		
It has decreased considerably	3%	
It has decreased slightly	1%	
It has remained the same	7%	
It has increased slightly	11%	
It has increased considerably	78%	
Have any of your COVID-19 patients passed away?		
No	8%	
Yes	92%	
Were any family members present when your patient/s were dying?		
No	71%	
Yes	29%	

**Table 2 ijerph-18-09243-t002:** Psychological and moral scales used.

Scales	*M* (SD)
**The Moral Distress Scale**	
Moral distress	2.5 (1.19)
**The Professional Quality of Life Scale**	
Compassion satisfaction	40.4 (5.5)
Burnout	27.5 (5.1)
Compassion fatigue	26.5 (6.2)
**The Professional Self-Care Scale**	
Physical self-care	3.79 (1)
Psychological self-care	2.81 (1.1)
Social self-care	3.97 (1)

**Table 3 ijerph-18-09243-t003:** Follow-up ANOVAs exploring differences in professional quality of life.

	Compassion Satisfaction					Burnout					Compassion Fatigue				
	*F*	df	Df Error	*p*	*η2*	*F*	df	Df Eror	*p*	*η2*	*F*	df	Df Error	*p*	*η2*
Sex	4.571	1	109	0.035	0.040	0.042	1	109	0.838	0.000	0.130	1	109	0.719	0.001
Occupation	0.276	2	107	0.759	0.005	0.636	2	107	0.531	0.012	3.021	2	107	0.053	0.053
Protective equipment against COVID-19	6.930	1	107	0.010	0.061	7.915	1	107	0.006	0.069	2.950	1	107	0.089	0.027
Providing patient support	6.856	1	100	0.010	0.064	1.612	1	100	0.207	0.016	2.744	1	100	0.101	0.027

**Table 4 ijerph-18-09243-t004:** Means and standard derivations for differences in professional quality of life.

Factors and Categories	Moral Stress		Compassion Satisfaction		Burnout		Compassion Fatigue	
	Mean	SD	Mean	SD	Mean	SD	Mean	SD
Sex								
Male	2.39	0.54	43.06	6.23	27.56	5.48	26.11	7.47
Female	2.54	0.75	40.15	5.08	27.29	4.95	26.70	6.09
Occupation								
Intensive care doctor	2.36	0.46	39.83	6.30	28.83	4.98	27.00	6.39
Nurse	2.55	0.67	40.55	5.24	27.12	4.74	25.68	5.87
Nursing Assistant	2.56	0.92	41.19	5.49	27.61	5.47	29.15	6.94
Protective equipment against COVID-19								
No	2.94	0.76	38.08	7.20	29.83	5.12	28.62	6.68
Yes	2.40	0.66	41.28	4.58	26.64	4.83	26.14	6.13
Providing patient support								
No	2.58	0.67	39.68	5.68	28.09	5.05	27.47	6.59
Yes	2.46	0.86	42.70	4.23	26.76	4.19	25.16	5.91

**Table 5 ijerph-18-09243-t005:** Demographic and occupational profiles of interviewees.

Participant	Occupation	Sex	Age	Years of Professional Experience	Years of Experience in ICU	Length of Time Providing COVID-19 Patient Care Prior to Interview
P1	Intensive care doctor	Male	53	28	19	3 months
P2	Intensive care doctor	Female	42	12	7	4 months
P3	Intensive care doctor	Male	54	21	20	4 months
P4	Intensive care doctor	Female	32	8	1.5	8 months
N1	Intensive care nurse	Male	44	23	20	4 months
N2	Intensive care nurse	Male	47	25	18	3 months
N3	Intensive care nurse	Female	46	25	20	3 months
N4	Intensive care nurse	Female	52	29	24	1 month
NA1	Nursing assistant	Male	30	8	18 months	7 months
NA2	Nursing assistant	Female	49	18	13	4 months
NA3	Nursing assistant	Female	43	20	20	8 months

**Table 6 ijerph-18-09243-t006:** Themes, subthemes, and representative quotes relating to the impact of COVID-19.

Themes and Subthemes	First Wave	Second Wave
**Clinical impact: Changes in clinical practice or in the hospital, such as workload, rearrangement of spaces, protective equipment, dehumanisation of care.**	You never think you’ll experience something like this. At first, I couldn’t believe it (P2) My first impression is that this has been a very intense, unforeseen, unexpected experience (N1) A flood of patients came in, all of them critically ill, one after another. We worked shifts under a lot of pressure, wearing PPE for many hours (N1) The operating theatres were not operational. The resuscitation unit was to serve as a regular ICU for non-COVID critical patients and then a space was made available for non-critical COVID patients (P2) They were all novice professionals, but they were eager to give their all. You had [to train] two of them at a time. It was an excessive workload (N4) The most difficult part was working with the PPE on, especially for nurses. They worked longer hours, and when they were finished, you could see the marks on their faces (P1) You’d put on your goggles, and they’d start to fog up. I’m sorry, but I chose to work at ease at the expense of my personal safety. I don’t know how other people could prepare medication with those screens on (N2) PPE and masks would change from day to day. Some masks were not protecting us (P2) Those of us in the ICU are the hospital elite. It was shameful to see how others were working in other departments. We didn’t have equipment shortages (NA3) It wasn’t humane, you couldn’t even hold their hand. It was like an assembly plant (NA3) We felt that the care we provided was becoming increasingly dehumanized, but our relationship with the families was excellent, everyone understood the situation (N1) We all felt as though we had kidnapped the people we had in the ICU (P2) It was hard seeing no relatives in the ICU, the patients alone, the deaths, informing the families by phone (N4) Seeing patients alone, intubated, and only able to say goodbye to us. The desolation felt by patients was brutal (N1) They were patients with multiple conditions, and you were extremely happy when they recovered (N3)	There’s more information, more patient circuits and protocols; you’re more prepared, precautionary measures are taken (N1) Activity has never stopped inside the hospital. And people from other departments can no longer come and help (NA1) There are not enough people to hire and not enough people who are familiar with the ICU. People are working double shifts. It’s exhausting (N1) New people came during the first wave, but now more new people have come again (NA3) There were no longer so many concerns about PPE (P4) During the second wave, we were more prepared and informed about unfamiliar things. We’re now more acquainted with PPE and the disease, among other things (N1)
**Professional impact: Changes that directly affected the professional, such as burnout, compassion satisfaction, and compassion fatigue.**	This is the first time I’ve seen colleagues seriously considering leaving the nursing profession (N4) It’s been very positive in allowing you to analyze and appreciate things (NA2) We all need to wear the word ‘humility’ on our foreheads [as a reminder] (N4) It was amazing how colleagues from different professions set to work together. It was terrific. Doctors, nurses, and assistants worked shoulder to shoulder. The hierarchical boundaries that still linger in our imaginations became increasingly blurred (N1)	People are more tired in this second wave. They haven’t been able to take a few days off or go on holiday (N1) Burnout undermines resilience. People are tired, physically and emotionally exhausted (NA3) So now people are tired... physically tired, emotionally tired (N1) We felt helpless and outraged during the phased reopening because we saw that measures were being lifted as if we had beaten the virus while we still had COVID patients in our units (P4) People are very tired. They won’t stop whining and complaining (NA3) They’re not giving us the psychological support we need. I myself am considering seeking professional help, which I believe our organizations should be offering us (P4)
**Personal and family impact: personal and family changes such as insomnia, emotional lability, and fear of infecting others.**	I’d wake up to my heart pounding at 5:00 in the morning and I’d start looking for solutions. It started to feel unhealthy and made you wonder what was going wrong (N1) It was chaos at the beginning. All that made us extremely nervous and stressed (NA2) You couldn’t cry when you were in there. You cried when you got out. In my case, I’d cry when I got home (NA3) The first time I was putting on my PPE, all I had in my head was the image of my children. I had fear written all over my face. I wasn’t afraid for myself, but for my children and my husband (N4) My ritual was like this: when my husband came to pick me up, I’d sit on a towel in the back of the car. I’d come home and rub bleach all over me, get completely undressed, put my clothes in the washing machine, and go straight to the shower. And then you could speak to me. I was sleeping on a mattress on the floor in the dining room, away from them (N4) It has affected my private life because I self-isolated at home in a room (P1)	You reduce your personal life and your leisure time to walking around and not interacting with anyone, and then you keep seeing that people are not observing that (N1) You remember every patient, especially those who didn’t turn out well. You have very painful memories, and these are things that we’ll never forget. They are engraved on our minds. For a year now I haven’t slept well, I have nightmares, my life at home has changed dramatically (P4)

## Data Availability

The datasets generated and/or analyzed during the current study will not be publicly available due to privacy and confidentiality reasons, but they will be available from the corresponding author upon reasonable request.

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
