# Peer review of "The Impact of the COVID-19 Pandemic on ICU Healthcare Professionals: A Mixed Methods Study"

_ijerph, 2021, doi:10.3390/ijerph18179243_

Round 1
Reviewer 1 Report
Évaluation “The impact of the COVID-19 pandemic on ICU healthcare professionals: a mixed methods study.”
International Journal of Environmental Research and Public Health.
This is, in my opinion, an interesting study examining the association between the provision of COVID-19 patient care on ICU healthcare professionals. The subject of research has a high social importance. This study has many strengths. First, it is a mixed methods study. Second, the results provide an advance in current knowledge. Third, the article is well written and structured. Data are presented appropriately. Lastly, these results should be of interest for the readership of the Journal. However, I have some comments and concerns that I hope will be helpful.
1-) A suggestion would be to add a theoretical background in order to support your model analysis and results beyond the empirical support.
2-) Although it is true that the mix method is a strength, several limitations of this study were not mentioned or elaborated.
- Additional covariates could have been included in the quantitative analysis, in line with previous research founding that several variables were associated with the outcomes (e.g., parental status, marital status, social support outside the workplace). It is possible that those covariates would explain workers’ mental health.
- The small sample should be discussed further. Not only in terms of the explanation of the low response rate, but also on the possible consequences on the results.
- The fact that the set of measurements came from the same source leads to the possibility of a common variance bias.
-Cross-sectional dataset for the quantitative results.
- The use of self-report questionnaire/quantitative part.
3-) Suggestions for future research should be added.
4-) Practical implications should be discussed.
5-) Considering this study limitation, this sentence should be modified on page 4/18: “... we may conclude that the perception of a safe environment acts as a protector against burnout syndrome and moral distress”. “Act as a protector” should be replace with “was associated with”.
I would like to congratulate the authors for the work !
Author Response
Response to Reviewer 1 Comments:
This is, in my opinion, an interesting study examining the association between the provision of COVID-19 patient care on ICU healthcare professionals. The subject of research has a high social importance. This study has many strengths. First, it is a mixed methods study. Second, the results provide an advance in current knowledge. Third, the article is well written and structured. Data are presented appropriately. Lastly, these results should be of interest for the readership of the Journal. However, I have some comments and concerns that I hope will be helpful.
Thank you very much for your input and your suggestions. Please find below the answers to the points raised.
Point 1: A suggestion would be to add a theoretical background in order to support your model analysis and results beyond the empirical support.
Response 1: Theoretical background on variables related to professionals’ quality of life and moral distress has been added.
Point 2: Although it is true that the mix method is a strength, several limitations of this study were not mentioned or elaborated.
- Additional covariates could have been included in the quantitative analysis, in line with previous research founding that several variables were associated with the outcomes (e.g., parental status, marital status, social support outside the workplace). It is possible that those covariates would explain workers’ mental health.
- The small sample should be discussed further. Not only in terms of the explanation of the low response rate, but also on the possible consequences on the results.
- The fact that the set of measurements came from the same source leads to the possibility of a common variance bias.
- Cross-sectional dataset for the quantitative results.
- The use of self-report questionnaire/quantitative part.
Response 2: We agree with the reviewer that these are limitations of our manuscript, and consequently, we have added them in the limitations section. Specifically, this section it now reads like this:
“The major weakness of this study is the small sample size, due to low response rate to the electronic survey, indicating particular difficulties in reaching the target sample via email. Possible explanations include the fact that not all employees use their work email addresses or that their workloads are so heavy that responding to a questionnaire is not viewed as a priority. This small sample size and response rate could affect the generalization of the results. Other shortcomings of current research include the absence of covariates that have been traditionally related to workers’ mental health, such as parental status, marital status, or social support outside the workplace; the fact that the set of measurements came from the same source, which could lead to the possibility of a common variance bias; the cross-sectional nature of the study; or the use of self-report measures for the quantitative part.”
Point 3: Suggestions for future research should be added. Practical implications should be discussed.
Response 3: The implications have been made more explicit, adding educational and research implications. In addition, future lines of research have been incorporated and discussed.
Point 4: Considering this study limitation, this sentence should be modified on page 4/18: “... we may conclude that the perception of a safe environment acts as a protector against burnout syndrome and moral distress”. “Act as a protector” should be replace with “was associated with”.
Response 4: Thank you very much for your appreciation, we have made the proposed change.
Reviewer 2 Report
I read with great interest the article entitled “The impact of the COVID-19 pandemic on ICU healthcare professionals: a mixed methods study.”
.”
In their work the authors try to explore the impact of the provision of COVID-19 patient care on ICU healthcare professionals by a self-report online survey based on three validated questionnaires, and individual semi-structured in-depth online interviews.
The article is well written but has some critical issues that must be corrected to make it publishable.
Major concerns.
Introduction:
Data on the general population are fine. I think it would be appropriate to introduce some data regarding the pandemic (infected and dead) among health workers to provide a clearer dimension of the problem. Authors could draw inspiration from published works concerning other countries [COVID-19 and Italian healthcare workers from the initial sacrifice to the mRNA vaccine: Pandemic chrono-history, epidemiological data, ethical dilemmas, and future challenges." Frontiers in Public Health 8 (2020)].
The authors speak of anxiety on the part of ICU health professionals. The phenomenon has been ubiquitous. I believe it is important to ask the authors to specify whether, as has happened in other states, the fear of legal action against doctors has also contributed to this phenomenon in their reality. (i.e COVID-19 and medical liability: Italy denies the shield to its heroes." EClinicalMedicine 25, 2020).
Methods:
In general, the chapter must be written more clearly. There must be clear methods of inclusion and selection, calculation of the number, etc. etc.
1) The three validated questionnaires for the sample of 122 individuals are fine. The type of questionnaire administered in the individual has semi-structured in-depth online interviews been validated?
2) The authors should specify how the calculation of the sample size was made. In particular, how was the 11-person sample selected? Has the estimate of the sample size been carried out?
3) The authors should clarify if the work was standardized on the basis of the educational qualification, sex, race / ethnicity of the sample.
Discussion:
The theme of variations in the various activities has been thoroughly investigated for different specialties [
The authors asked the study subjects whether in their opinion activity had increased or decreased. I think that the data can also be obtained objectively in a reality such as that of the Balearics, it would be interesting to provide this information also during the discussion. [Fear of the COVID-19 and medical liability. Insights from a series of 130 consecutives medico-legal claims evaluated in a single institution during SARS-CoV-2-related pandemic." (2021).
Minor concerns
I ask the authors to expand the literature on the subject comparing their data with that reported by other studies. (Aprato, Alessandro, et al. "Clinical Activities, Contaminations of Surgeons and Cooperation with Health Authorities in 14 Orthopedic Departments in North Italy during the Most Acute Phase of Covid-19 Pandemic." International Journal of Environmental Research and Public Health 18.10 (2021): 5340.; Larribère, Lionel, et al. "Assessment of SARS-CoV-2 Infection among Healthcare Workers of a German COVID-19 Treatment Center." International Journal of Environmental Research and Public Health 18.13 (2021): 7057.; Kim, Min-Young, and Yun-Yi Yang. "Mental Health Status and Its Influencing Factors: The Case of Nurses Working in COVID-19 Hospitals in South Korea." International Journal of Environmental Research and Public Health 18.12 (2021): 6531.; Tahara, Masatoshi, Yuki Mashizume, and Kayoko Takahashi. "Mental Health Crisis and Stress Coping among Healthcare College Students Momentarily Displaced from Their Campus Community Because of COVID-19 Restrictions in Japan." International Journal of Environmental Research and Public Health 18.14 (2021): 7245.; Bisesti, Alberto, et al. "Facing COVID-19 Pandemic in a Tertiary Hospital in Milan: Prevalence of Burnout in Nursing Staff Working in Sub-Intensive Care Units." International Journal of Environmental Research and Public Health 18.13 (2021): 6684.]
I believe that authors should meet at least the major concerns required to make the article suitable for publication
Author Response
Response to Reviewer 2 Comments:
I read with great interest the article entitled “The impact of the COVID-19 pandemic on ICU healthcare professionals: a mixed methods study.”
In their work the authors try to explore the impact of the provision of COVID-19 patient care on ICU healthcare professionals by a self-report online survey based on three validated questionnaires, and individual semi-structured in-depth online interviews.
The article is well written but has some critical issues that must be corrected to make it publishable.
Thank you very much for your suggestions. Please find below the answers to the points raised.
Point 1: Introduction:
- Data on the general population are fine. I think it would be appropriate to introduce some data regarding the pandemic (infected and dead) among health workers to provide a clearer dimension of the problem. Authors could draw inspiration from published works concerning other countries [COVID-19 and Italian healthcare workers from the initial sacrifice to the mRNA vaccine: Pandemic chrono-history, epidemiological data, ethical dilemmas, and future challenges." Frontiers in Public Health 8 (2020)].
Response 1.1: We have introduced some data regarding the pandemic among health workers in Spain based on the data available in our country.
- The authors speak of anxiety on the part of ICU health professionals. The phenomenon has been ubiquitous. I believe it is important to ask the authors to specify whether, as has happened in other states, the fear of legal action against doctors has also contributed to this phenomenon in their reality. (i.e COVID-19 and medical liability: Italy denies the shield to its heroes." EClinicalMedicine 25, 2020).
Response 1.2.: In Spain, fear of legal claims has not been described as a cause of anxiety for intensive care units’ professionals. In fact, in the interviews, the professionals consider that, in our community, they have not faced ethical conflicts other than the usual pre-COVID practice, especially related to the limitation of admission to the ICU or of therapeutic effort. However, we have revised the proposed article and we have included it in the discussion section.
Point 2: Methods: In general, the chapter must be written more clearly. There must be clear methods of inclusion and selection, calculation of the number, etc. etc.
- The three validated questionnaires for the sample of 122 individuals are fine. The type of questionnaire administered in the individual has semi-structured in-depth online interviews been validated?
Response 2.1.: The semi-structured in-depth online interviews conducted in this study were not validated. In qualitative research, qualitative interviews are not instruments subject to validation, unlike questionnaires administered in quantitative research. However, as a rigorous strategy, pre-interviews were conducted prior to the formal interviews to ensure the rationality of the interview structure and the representativeness of the subjects. This allowed us to check the participants' understanding of the questions included in the interview, to correct possible ambiguities, to make a first approach to the different analysis topics that emerged through them and to verify, ultimately, that the study objectives could be adequately explored through the proposed questions. More information on the process of piloting qualitative interviews can be found in: "Piloting interviews in qualitative research: operationalization and lessons learned". International Journal of Academic Research in Business and Social Sciences (2017): 1073-1080. This article has been added in materials and methods section.
- The authors should specify how the calculation of the sample size was made. In particular, how was the 11-person sample selected? Has the estimate of the sample size been carried out?
Response 2.2.: In qualitative research, the determination of the sample size is contextual and depends in part on the scientific paradigm under which the research is conducted. Qualitative research uses statistically non-representative samples. The sample size is not determined by the need to ensure generalisability, but by a desire to investigate fully the chosen topic and provide information-rich data (Corbich 1999). Therefore, they are small samples where data saturation is sought as a principle of rigor.
International Journal for Quality in Health Care; Volume 19, Number 6: pp. 349 –357 10.1093/intqhc/mzm042 Advance Access Publication: 14 September 2007
Consolidated criteria for reporting qualitative research (COREQ): a 32-item checklist for interviews and focus groups ALLISON TONG1,2, PETER SAINSBURY1,3 AND JONATHAN CRAIG
- The authors should clarify if the work was standardized on the basis of the educational qualification, sex, race / ethnicity of the sample.
Response 2.3: Participant profiles were devised based on profession and gender. The sample was balanced in proportion to the number of professionals in each ICU. Sociodemographic variables that have been taken into account are shown in table 5 -workplace is not explained in the table to guarantee the anonymity of the participants-.
Point 3: Discussion: The theme of variations in the various activities has been thoroughly investigated for different specialties [
The authors asked the study subjects whether in their opinion activity had increased or decreased. I think that the data can also be obtained objectively in a reality such as that of the Balearics, it would be interesting to provide this information also during the discussion.
[Fear of the COVID-19 and medical liability. Insights from a series of 130 consecutives medico-legal claims evaluated in a single institution during SARS-CoV-2-related pandemic." (2021).
Response 3: Unfortunately, there is no published bibliography in relation to these indicators of the Balearic hospitals for this period of time. In the introduction we made reference to the information available by the Balearic Health Service.
Point 4: Minor concerns. I ask the authors to expand the literature on the subject comparing their data with that reported by other studies.
(Aprato, Alessandro, et al. "Clinical Activities, Contaminations of Surgeons and Cooperation with Health Authorities in 14 Orthopedic Departments in North Italy during the Most Acute Phase of Covid-19 Pandemic." International Journal of Environmental Research and Public Health 18.10 (2021): 5340.; Larribère, Lionel, et al. "Assessment of SARS-CoV-2 Infection among Healthcare Workers of a German COVID-19 Treatment Center." International Journal of Environmental Research and Public Health 18.13 (2021): 7057.; Kim, Min-Young, and Yun-Yi Yang. "Mental Health Status and Its Influencing Factors: The Case of Nurses Working in COVID-19 Hospitals in South Korea." International Journal of Environmental Research and Public Health 18.12 (2021): 6531.; Tahara, Masatoshi, Yuki Mashizume, and Kayoko Takahashi. "Mental Health Crisis and Stress Coping among Healthcare College Students Momentarily Displaced from Their Campus Community Because of COVID-19 Restrictions in Japan." International Journal of Environmental Research and Public Health 18.14 (2021): 7245.; Bisesti, Alberto, et al. "Facing COVID-19 Pandemic in a Tertiary Hospital in Milan: Prevalence of Burnout in Nursing Staff Working in Sub-Intensive Care Units." International Journal of Environmental Research and Public Health 18.13 (2021): 6684.]
Response 4: Thank you very much for the suggestions on the literature review. We have reviewed the suggested articles and incorporated them in the introduction and/or discussion those that we have found suitable for this work:
- Bisesti, Alberto, et al. "Facing COVID-19 Pandemic in a Tertiary Hospital in Milan: Prevalence of Burnout in Nursing Staff Working in Sub-Intensive Care Units." International Journal of Environmental Research and Public Health 18.13 (2021): 6684.]
- Aprato, Alessandro, et al. "Clinical Activities, Contaminations of Surgeons and Cooperation with Health Authorities in 14 Orthopedic Departments in North Italy during the Most Acute Phase of Covid-19 Pandemic." International Journal of Environmental Research and Public Health 18.10 (2021): 5340
- Larribère, Lionel, et al. "Assessment of SARS-CoV-2 Infection among Healthcare Workers of a German COVID-19 Treatment Center." International Journal of Environmental Research and Public Health 18.13 (2021): 7057
- Kim, Min-Young, and Yun-Yi Yang. "Mental Health Status and Its Influencing Factors: The Case of Nurses Working in COVID-19 Hospitals in South Korea." International Journal of Environmental Research and Public Health 18.12 (2021): 6531.
Reviewer 3 Report
The topic of this manuscript was interesting and eye-catching. Some writing and methodological revisions would help to make the manuscript in its scientific soundness:
The introduction part needs further elaboration. From the design, moral distress appears as a very important factor influencing healthcare professionals’ professional quality of life. However, in the introduction part, moral distress was not mentioned. The significance of including moral distress should be shaped in the introduction section.
In data analysis part:
As there is no hypothesis ahead, the purposes of data analyses were not clear. Main sentences for each paragraph were required to guide the data analyses for a better readership.
Several mistakes were found in data analysis. On p.5, in lines 159-160, it should be “there is no evidence indicating significant differences in moral distress between two genders”. No significance ONLY indicates no evidence for the differences. Similarly, in lines 180-181, non-significance was misinterpreted for no effect. Please do a throughout check of such mistakes in the data analysis section.
The qualitative analyses part needs a re-shape. Please follow the standard of qualitative study reporting, such as reporting the method of how the themes were found, and how many researchers were coding independently, etc. Possible references please see:
O’Brien, B. C., Harris, I. B., Beckman, T. J., Reed, D. A., & Cook, D. A. (2014). Standards for reporting qualitative research: a synthesis of recommendations. Academic Medicine, 89(9), 1245-1251.
The discussion looks smooth. Please link the literature back to elaborate your introduction part.
Author Response
Response to Reviewer 3 Comments:
The topic of this manuscript was interesting and eye-catching. Some writing and methodological revisions would help to make the manuscript in its scientific soundness:
Thank you very much for your input, which we believe will result in the improvement of the manuscript. Please find below the answers to the points raised.
Point 1: The introduction part needs further elaboration. From the design, moral distress appears as a very important factor influencing healthcare professionals’ professional quality of life. However, in the introduction part, moral distress was not mentioned. The significance of including moral distress should be shaped in the introduction section.
Response 1: Theoretical background related to moral distress has been added in the introduction section.
Point 2: In data analysis part:
- As there is no hypothesis ahead, the purposes of data analyses were not clear. Main sentences for each paragraph were required to guide the data analyses for a better readership.
- Several mistakes were found in data analysis. On p.5, in lines 159-160, it should be “there is no evidence indicating significant differences in moral distress between two genders”. No significance ONLY indicates no evidence for the differences. Similarly, in lines 180-181, non-significance was misinterpreted for no effect. Please do a throughout check of such mistakes in the data analysis section.
Response 2.1 and 2.2: Subheadings and additional sentences were added for the quantitative analysis part of the results. Also, mistakes in this part pointed by the reviewer have been corrected. We have also checked the data analysis section.
- The qualitative analyses part needs a re-shape. Please follow the standard of qualitative study reporting, such as reporting the method of how the themes were found, and how many researchers were coding independently, etc. Possible references please see: O’Brien, B. C., Harris, I. B., Beckman, T. J., Reed, D. A., & Cook, D. A. (2014). Standards for reporting qualitative research: a synthesis of recommendations. Academic Medicine, 89(9), 1245-1251.
Response 2. The analysis part -specially coding and triangulation process- has been better explained. The role of the main researcher and the observers during the interviews has also been explained in the text of the article. The following text has been added to the article:
“Each interview and field notes log were codified independently by four researchers. Once finished, both researchers met to compare their results. Where their codifications differed, researchers explained their thinking processes. Through a process of dialogue and comparison, they reached an agreement on the coding system. Therefore, the codification of each interview and observation notes was the result of five codifications: four independent ones and a joint one. Once the list of codes had been completed, two team members drew up the analysis subcategories and categories and revised the codes under each of them. Afterwards, they compared the coherence of each code and revised the list of codes, eliminating some that lacked a specific sense and unifying the ones that, although meaning the same, had been codified with different names. It is important to note that few codes referring to different categories were unified and that most of the fusions of codes were produced in the same categories.”
Point 3: The discussion looks smooth. Please link the literature back to elaborate your introduction part.
Response 3: Thank you very much for the recommendation. The introduction and discussion literature have been reviewed.
Round 2
Reviewer 1 Report
The authors The authors have responded well to the various points raised. have responded well to the various points raised. Congratulations on your work !
Author Response
Thank you very much
Reviewer 2 Report
I read the authors' review. Unfortunately, strong doubts remain about the possibility to consider the article in its current form.
Assuming that they want to give constructive advice, if the Editor wishes to consider the paper for publication, the authors should add a chapter before the conclusions on the limitations of the study.
There are still strong doubts about the small sample number. [Vasileiou, Konstantina, et al. "Characterising and justifying sample size sufficiency in interview-based studies: systematic analysis of qualitative health research over a 15-year period." BMC medical research methodology 18.1 (2018): 1-18.].
There are also strong doubts about the use of non-validated questionnaires.
The lack of data on ethnicity and specialties should also be introduced between the limits or between future purposes for a future study.
The number of references is not yet sufficient to argue what is reported.
The article still requires some work to be considered for publication.
Author Response
Response to Reviewer 2 Comments:
I read the authors' review. Unfortunately, strong doubts remain about the possibility to consider the article in its current form.
Point 1. Assuming that they want to give constructive advice, if the Editor wishes to consider the paper for publication, the authors should add a chapter before the conclusions on the limitations of the study.
Response point 1:
Thank you very much for the recommendation related to making limitations visible. We have included a specific section before the conclusions.
First, the limitations of quantitative design have been explained because of the first review and then some of the limitations of qualitative design, especially those related to conversational techniques (on-line) versus naturalistic techniques.
Point 2: There are still strong doubts about the small sample number. [Vasileiou, Konstantina, et al. "Characterising and justifying sample size sufficiency in interview-based studies: systematic analysis of qualitative health research over a 15-year period." BMC medical research methodology 18.1 (2018): 1-18.].
Response Point 2:
In relation to the doubts about the number of the sample, we once again substantiate that there was saturation of the data obtained from the interviews carried out, being able to answer the objectives of the study. As Konstantina (2018) points out, a bibliographic reference that you suggested to us, the researchers' knowledge of the intensive care environment assured us of the quality and richness of the data, together with their experience in conducting qualitative interviews. In addition, we attach a paper by another author where he also points out that the sample size is contextual to the phenomenon and study design, and the importance of saturation as an element of rigor (Boddy CR. Sample size for Qualitative Interviews. Qual Mark Res An Int J. 2015; 19 (2003): 426–32).
On the other hand, as Konstantina (2018) points out, we have handled an initial verbatines table in which we had numerous expressions of each of the codes that guaranteed their saturation and the generation of the categories described. Finally, in the article, given the average length of this type of document, and due to the fact that it does not provide the reader with higher quality information, we have selected the most representative ones.
If you or the editor want to have the original table with all the verbatims identified, we could provide it, although this would mean translating a large number of verbatims into English.
Point 3: There are also strong doubts about the use of non-validated questionnaires
Response Point 3:
We would like to be able to solve the doubts of the reviewer. In this sense, we cite one of the most relevant Handbooks in qualitative health research when it refers to how interviews should be (Kelly S. Qualitative Interviewing Techniques and Styles. In: The SAGE Handbook of Qualitative Methods in Health Research [Internet]. 2010 . p. 307–8. Available from: https://books.google.es/books?hl=es&lr=&id=sGFn6q-VBeAC&oi=fnd&pg=PA307&ots=B1XvWbKFIs&sig=1vbU0BqIN1iPGSeKazixT9344O8&qyv_escfalse#T9344O8&qredv_escfalse
"The term" qualitative interviews "refers to interview techniques that provide qualitative (textually rich) data. A qualitative interview is unlike a structured or standardized interview, where the goal is to generate data amenable to quantitative analysis.
Informal interviews in an ethnographic field setting and "natural conversation" are at one end of the continuum and standardized interviews at the opposite end. Standardized interviews are highly structured in terms of question wording, order, and response categories (most questions being “fixed choice”) and are conducted in as standardized a manner as possible, with the researcher conducting the interview acting as a “neutral” instrument.
Qualitative interviews, on the other hand, explicitly involve the interviewer and respondent in interactions, as interaction partners. The data produced in qualitative interviews is understood explicitly to be the product of such interactions, with the attention to reflexivity and subjectivity that this involves. "
The question regarding interview scripts in qualitative research can also be consulted in other bibliography such as:
- DiCicco-Bloom B, Crabtree BF. The qualitative research interview. Med Educ. 2006; 40 (4): 314–21.
- Kallio H, Pietilä AM, Johnson M, Kangasniemi M. Systematic methodological review: developing a framework for a qualitative semi-structured interview guide. J Adv Nurs. 2016; 72 (12): 2954–65.
Point 4: The lack of data on ethnicity and specialties should also be introduced between the limits or between future purposes for a future study.
Response Point 4:
Ethnicity has not been included in the table since it has not been studied as a variable. However, we would like to clarify to you and the reviewer, that all participants are Caucasian, of Spanish nationality. In our Autonomous Community the differences by ethnicity among health professionals are very minority.
In relation to the specialty, possibly we had not explained well the reality of intensive care professionals in Spain. In this sense, we have made the specialty of doctors and nurses better visible in the article and in Table 5: Demographic and occupational profiles of interviews and we make the following clarifications.
- In Spain, there is an intensivist specialty for doctors. All the participants were. We have clarified it in the tables and in the descriptions.
- In relation to nurses, there is currently no recognized specialty for critical care nurses. This is a request that because of the pandemic is gaining momentum; However, when a nurse has been working in a special service for more than 4 years and undergoing specific ongoing training, in hospital contracts she is considered an expert in that service. Thus, the nurses interviewed can respond to what is considered an intensive care nurse in Spain.
- In relation to nursing assistants, there are no specialties by academic qualification.
We appreciate the proposal for future research. Our intention is to repeat the interviews in the coming months, one year later and compare with other settings. In this sense, the following text has been introduced (484-490):
“Finally, we consider it is important to point out some future research directions such as continuing to explore the differences and similarities of professionals' clinical, professional and personal impact between different health services (such as emergency services, primary health care services), between genders and between years of experience (especially of those healthcare students who started working in the context of the pandemic), and also to explore the short and long-term impact of the pandemic on patients and their family members, especially on chronically ill patients and high dependency patients ”.
Point 5: The number of references is not yet sufficient to argue what is reported
Response Point 5:
We appreciate your comment. In the first review we incorporated more bibliography into the discussion. However, we have reviewed the references proposed in the first review and have incorporated all the references in the text, mainly Nioi (2021). Thank you very much for the recommendation.
The article still requires some work to be considered for publication.
No doubt the comments have improved the paper. We are grateful to you. We hope that the modifications made are appropriate and we are at your disposal for whatever you consider.
Reviewer 3 Report
The manuscript has been largely improved.
Some minor mistakes should be further checked. For example, in line 189, followed "four researchers", the use of "both researchers" is grammatically problematic. Please include a professional copy-editing service to thoroughly check the manuscript before publication.
Author Response
Response to Reviewer 3 Comments:
The manuscript has been largely improved.
Thank you very much.
Point 1: Some minor mistakes should be further checked. For example, in line 189, followed "four researchers", the use of "both researchers" is grammatically problematic. Please include a professional copy-editing service to thoroughly check the manuscript before publication.
Response Point 1:
Thank you very much for identifying the error. We have revised the quoted line.
If the editor and reviewers deem it necessary, we will ask our certified translator to review the modifications made, and in this way we can attach the certificate for this second linguistic review.